# Site-Directed Mutations at Phosphorylation Sites in *Zea mays* PHO1 Reveal Modulation of Enzymatic Activity by Phosphorylation at S566 in the L80 Region

**DOI:** 10.3390/plants12183205

**Published:** 2023-09-08

**Authors:** Noman Shoaib, Nishbah Mughal, Lun Liu, Ali Raza, Leiyang Shen, Guowu Yu

**Affiliations:** 1State Key Laboratory of Crop Gene Exploration and Utilization in Southwest China, Sichuan Agricultural University, Chengdu 611130, China; noman@cib.ac.cn (N.S.); nishbahmughal97@gmail.com (N.M.); liulun@stu.sicau.edu.cn (L.L.); S20166005@stu.sicau.edu.cn (L.S.); 2National Demonstration Center for Experimental Crop Science Education, College of Agronomy, Sichuan Agricultural University, Chengdu 611130, China; 3CAS Key Laboratory of Mountain Ecological Restoration and Bioresource Utilization & Ecological Restoration Biodiversity Conservation Key Laboratory of Sichuan Province, Chengdu Institute of Biology, Chinese Academy of Sciences, Chengdu 610041, China; aliraza@cib.ac.cn; 4University of Chinese Academy of Sciences, Beijing 101408, China

**Keywords:** activity, subcellular localization, plastidial starch phosphorylase, phosphorylation

## Abstract

Starch phosphorylase (PHO) is a pivotal enzyme within the GT35-glycogen–phosphorylase (GT; glycosyltransferases) superfamily. Despite the ongoing debate surrounding the precise role of PHO1, evidence points to its substantial influence on starch biosynthesis, supported by its gene expression profile and subcellular localization. Key to PHO1 function is the enzymatic regulation via phosphorylation; a myriad of such modification sites has been unveiled in model crops. However, the functional implications of these sites remain to be elucidated. In this study, we utilized site-directed mutagenesis on the phosphorylation sites of *Zea mays* PHO1, replacing serine residues with alanine, glutamic acid, and aspartic acid, to discern the effects of phosphorylation. Our findings indicate that phosphorylation exerts no impact on the stability or localization of PHO1. Nonetheless, our enzymatic assays unveiled a crucial role for phosphorylation at the S566 residue within the L80 region of the PHO1 structure, suggesting a potential modulation or enhancement of PHO1 activity. These data advance our understanding of starch biosynthesis regulation and present potential targets for crop yield optimization.

## 1. Introduction

Phosphorylase (EC 2.4.1.1), discovered in 1939, is instrumental in the reversible transfer of glucosyl units from glucose-1-phosphate (Glc-1-P) to glycogen, enabling the liberation of inorganic phosphates [1]. This enzyme, initially identified in mammalian liver tissue, is now recognized across various life forms including animals, plants, algae, and bacteria, emphasizing their evolutionary connections [2]. In a divergence from animals, higher plants and green algae host two distinctive starch phosphorylase variants, the plastidic form (PHO1 or SP-L) and the cytoplasmic form (PHO2 or SP-H), each differing in molecular size, substrate specificity, and physiological functions [3,4].

Notably, plant phosphorylases bear structural similarities to animal and bacterial counterparts, like maltodextrin phosphorylase, yet they are distinguished by a unique N-terminus and an additional peptide segment known as the L80 domain [5,6,7,8]. This L80 domain, exclusively present in PHO1, is thought to confer the diversified biochemical activities of the enzyme [9,10]. It encompasses a variable set of negatively charged amino acids, phosphorylation sites, and a putative PEST motif [2,11]. The L80 segment in potatoes reportedly obstructs polysaccharide binding to PHO1, presumably due to steric hindrance [12]. Moreover, in the *Ipomoea batatas*, when its N-terminal amino acid residues are phosphorylated, it serves as an active substrate for the 20S or 26S proteasome [13]. These insights lay the foundation for further investigation into the functional gradations of different phosphorylase forms across species.

The expression patterns and subcellular localization of PHO1, particularly within plastids, underscore its significant role in starch biosynthesis, as evidenced by correlations with several starch biosynthetic enzymes [8,14]. Starch biosynthesis is a multifaceted process engaging over 15 enzymes, including starch synthases (SSs, EC 2.4.1.21), starch branching enzymes (SBEs, EC 2.4.1.18), and starch de-branching enzymes (DBEs, 3.2.1.41; predominantly isoamylases (ISAs)). Each enzyme has tissue- and development-specific isoforms [15]. PHO1 may play an indispensable role during the early stages of endosperm development, as demonstrated by its higher activity than SSs during initial maize kernel development stages [16]. The biochemical evidence suggest that PHO1 acts on glucans to reversibly produce Glc-1-P and Pi [11]. In the synthetic pathway, PHO1 facilitates the transfer of glucosyl units from Glc-1-P to expanding glucan chains, releasing Pi. In contrast, the phosphorolytic route sees the same reaction in the reverse direction [17,18,19]. The activity of PHO1 in starch biosynthesis is dictated by the Pi/Glc-1-P ratio, with higher ratios favoring glucan synthesis, and lower ratios favoring degradation [20]. However, factors beyond substrate availability can also steer the direction of PHO1 activity [21]. This understanding adds an essential layer of complexity to the regulation of starch biosynthesis, warranting further exploration into the factors controlling PHO1 activity.

The phosphorylation-dependent regulation of major starch biosynthetic enzymes was first elucidated in wheat plants, highlighting the phosphorylation of serine residues in SBEs isoforms (SBEI, SBEIIa, and SBEIIb) [22,23]. Phosphorylation stimulated SBEIIa activity, while dephosphorylation attenuated it. Similarly, PHO1 phosphorylation has been identified as a crucial element during starch filling, with its regulation via protein phosphorylation observed across various model crops including *Zea mays*, *Oryza sativa*, *Hordeum vulgare*, and *Triticum aestivum* [23,24,25,26,27]. Moreover, the phosphorylation of the brittle 2 (Bt2) subunit of ADP-glucose pyrophosphorylase (AGPase) was found to boost its activity in *Zea mays* [28]. In animal systems, gPHO was revealed to coordinate with glycogen synthase (GS), necessitating gPHO phosphorylation for the activation of glycogen synthase, thereby regulating glycogen synthesis and degradation [29]. Structural alterations in gPHO at the subunit interface are triggered by conformational transitions at the amino terminus upon protein phosphorylation, resulting in a homodimer structure of gPHO1 [30].

To elucidate the role of phosphorylation in *Zea mays* PHO1, phosphorylation sites have been identified, including serine 69, threonine 537, serine 547, serine 556, serine 565, and serine 566. To decipher the impact of phosphorylation on PHO1 activity, stability, and subcellular localization, site-directed point mutations were executed at serine 69 and serine 566, and recombinant PHO1 was generated. The findings suggest that phosphorylation at serine 566 may primarily modulate PHO1 activity, while the stability and subcellular localization appear unimpacted by phosphorylation.

## 2. Results

### 2.1. Antibody Preparation and Specificity Detection

For the protein expression and purification, the PGEX vector system was employed. The PGEX-6T vector allows for the expression of fusion proteins tagged with Glutathione S-transferase (GST), facilitating easy purification. GST is a widely used tag due to its robust expression and ease of purification using glutathione-based resins. The 651 bp fragment from the C-terminal of PHO1 (PHO1-C) gene was amplified (Figure 1C) and cloned to PGEX-6T vector to generate PGEX-6T-PHO1-C plasmid. The PGEX-6T-PHO1-C expression vector was successfully constructed by double-digesting the recombinant plasmid PGEX-6T-PHO1-C with BamHI and EcoRI, obtaining a fragment with a band size of 651 bp, which was consistent with the expected results (Figure 1D). Further sequencing of the positive clones of the recombinant plasmid confirmed the successful construction of the PGEX-6T-PHO1-C expression vector.

The constructed prokaryotic expression vector PGEX-6T-PHO1-C was transformed into the *E. coli* BL21 (DE3) strain, which was then induced with 0.5 mmol/L IPTG. After being identified via 10% SDS-PAGE, a significant number of recombinant PHO1 protein bands were obtained from 48 kDa–55 kDa, consistent with the expected theoretical molecular weight (about 48 kDa; calculated based on its amino acid sequence, adjusting for the weight of water released during peptide bond formation, using the ExPASy ProtParam tool) (Figure 1E). The concentration of the recombinant protein was measured and compared to that of the BSA control, which showed that the three concentrations of the recombinant protein GST-PHO1-C were significantly higher than those of the BSA control. These results indicated that the GST-PHO1-C recombinant protein was successfully purified and could be used as an antigen for the preparation of an antibody. To investigate the PHO1 antibody specificity, the total protein was extracted from maize seeds in several developmental stages, and bands of similar size were detected using Western blotting at approximately 112 kDa (Figure 1G), consistent with the predicted PHO1 size in maize.

### 2.2. PHO1 Expression and Distribution Pattern

#### 2.2.1. Transcripts of PHO1 Expression and Distribution Pattern in Developing Seeds

Our observations, acquired through real-time quantitative polymerase chain reaction (RT-qPCR) analysis, point towards the complex expression pattern of PHO1 in *Zea mays*, differing notably from other model crops. Particularly, PHO1 expression presented a dynamic landscape, being significantly amplified in the endosperm, while diminished in the anthers (Appendix A). Intriguingly, this pattern aligns with key temporal shifts during seed development, with PHO1 expression being initially subdued, escalating to a peak at 12 DAP, and subsequently waning as seed development reaches completion. Notably, the expression dynamics of PHO1 transcripts displayed a similar trajectory to that of various enzymes integral to starch biosynthesis, including SSs, SBEs, and DBEs [2,31]. This further emphasizes the possible role of PHO1 in the regulation of the starch biosynthesis pathway in *Zea mays*.

#### 2.2.2. Protein (PHO1) Expression and Distribution Pattern in Developing Seeds

The total protein was extracted (Appendix A) and through Western blot analysis, we charted the temporal profile of PHO1 expression in *Zea mays*, observing its presence across all developmental stages from the first DAP onwards. Our data depict a progressive increase in protein expression, reaching its peak between 16 to 22 DAP, before initiating a gradual decline until 26 DAP. However, this decline is punctuated by an unexpected surge in PHO1 expression during the later stages of seed development (Appendix A).

Remarkably, the pattern of PHO1 expression mirrors that of several starch biosynthesis-related enzymes, including SSs, SBEs, DPEs, and SBEs, thus pointing towards its potential role in the regulation or execution of starch biosynthesis in *Zea mays*. Comparative analysis with other model crops revealed that the PHO1 expression trajectory in *Zea mays* resonates with that of Hordeum vulgare, suggesting potential conservation of response across these crop species.

#### 2.2.3. Phos-Tag^TM^ Enrichment of PHO1 Phosphorylated Proteins

Maize endosperm proteins extracted from the developmental stages 15, 20, and 25 DAP were subjected to Phos-tag™ enrichment to isolate phosphorylated proteins. After the removal of non-specifically bound proteins via water washing, Western blotting was carried out using an PHO1-specific antibody to identify PHO1 in the enriched phosphorylated protein sample. Even though Phos-tag™ does not perfectly enrich all phosphorylated proteins, clear bands corresponding to PHO1 proteins were evident. Upon treatment of identical maize endosperm protein samples with alkaline phosphatase, these PHO1 bands either disappeared or were significantly weakened (Figure 2). These findings suggest that PHO1 likely undergoes phosphorylation during the process of starch biosynthesis.

#### 2.2.4. Phosphorylation Sites Prediction

Bioinformatics analysis was performed to predict potential phosphorylation sites in the N-terminus and L80 domain of the *Zea mays* PHO1 protein sequence. Serine, threonine, and tyrosine residues were found to be the commonly phosphorylated amino acid residues. The results indicated that multiple sites within both peptides could potentially be phosphorylated, with serine being the most likely target, followed by threonine. The predicted phosphorylation sites were assigned scores based on their probability of phosphorylation, with scores above 0.5 being considered highly likely, while scores between 0.3 and 0.4 were also presented in Appendix A. The N-terminus and L80 domain of *Zea mays* PHO1 were found to have several potential phosphorylation sites with scores higher than 0.5.

#### 2.2.5. iTRAQ (Isobaric Tags for Relative and Absolute Quantitation) Identification of ZmPHO1 Phosphorylation Sites

We utilized Phos-tag^TM^ technology to enrich phosphorylated proteins from endosperm samples collected at different stages of maize pollination and identified a large number of phosphorylated proteins involved in the starch synthesis pathway through mass spectrometry analysis. Based on these findings, we speculated that ZmPHO1 may be phosphorylated by kinases. Further analysis using the iTRAQ revealed two specific phosphorylation sites, namely Serine 69 in the N-terminal peptide and Serine 566 in the L80 domain of PHO1 (Table 1, Figure 3).

#### 2.2.6. Conservation of Identified Phosphorylation Sites in Monocots and Dicots

Multiple alignments were performed to determine the conservation of S69 and S566 phosphorylation sites in various model crops from monocots and dicots. The results showed that the serine in both domains was highly conserved only in monocot crops. While dicot crops had aspartic acid and glutamic acid instead of serine in the N-terminus, no serine, aspartic acid, and glutamic acid was present on S566 (Appendix A).

### 2.3. Site-Directed Point Mutations

The coding sequence of the PHO1 gene was amplified from maize endosperm 20d cDNA, resulting in a single specific band of approximately 2955 bp (Appendix A). Sequencing and PCR results confirmed that the purified target sequence was successfully inserted into the pMD-19T vector (Appendix A). To generate site-directed point mutations on S69 and S566 of PHO1, PHO1-19t plasmid was used. The serine coding base pairs at the respective amino acid positions were replaced with alanine, glutamic acid, or aspartic acid coding base pairs using a mutation method described in the material and method section. After mutation PCR and competent cell transfer, successful clones were generated and sequenced to verify the mutations. S566A, S566E, and S566D were easily generated, while S69A and S69E were also successfully generated, but S69D could not be achieved, possibly due to higher GC content. The successful and verified mutation site sequences are provided in Appendix A. The mutated clones were then used to amplify the mutated PHO1 and recombine with various required vectors.

### 2.4. Subcellular Localization and Stability of ZmPHO1

Subcellular localization refers to the specific presence of a protein or its expression product within a cell. Accurate subcellular localization of mature proteins is crucial for their stable biological functions and protein interactions are often confined to the same or closely located sub-organelles. The green fluorescent protein (GFP) reporter gene is a useful tool for determining protein location, by fusing the target gene and the GFP reporter gene. The proteins’ location can be accurately determined by observing the green fluorescence emitted under the laser irradiation of a fluorescence microscope. In this study, a PHO1-eGFP fusion protein vector was constructed (pCAMBIA-2300: PHO1-eGFP) and successfully expressed after transformation. The constructed plasmids containing the mutated PHO1-19t were recombined with the PHO1-eGFP vector to determine the subcellular localization of the mutant proteins. The stability of the mutated PHO1 proteins was not significantly affected. The results showed that the 35S-driven ZmPHO1 and eGFP fusion protein was specifically expressed in the amyloplasts of maize endosperm protoplasts (Figure 4). However, the green fluorescence signals of the S69A and S566A mutants, which simulated dephosphorylation, were more evenly distributed throughout the cells. On the other hand, the fluorescence signals of the phosphorylation-mimicking mutants S69E and S566E were similar to those of the ZmPHO1 wild-type protein, with the fluorescent signals focused on the amyloplasts. These findings suggest that the phosphorylation of ZmPHO1 at S69 and S566 may or may not affect its subcellular localization in maize endosperm protoplasts.

### 2.5. Activity of PHO1

#### 2.5.1. Expression and Purification of Recombinant PHO1

The expression of the cloned genes was assessed using SDS-PAGE and Western blot techniques. Both non-mutated and mutated ZmPHO1 were expressed by inducing with 1 mM IPTG, as described in the section Materials and Methods. Soluble extracts were collected from the induced and non-induced control groups to analyze the expression of recombinant ZmPHO1. The molecular weight of the ZmPHO1 protein fused with the His label was around 120 kDa. SDS-PAGE electrophoresis revealed clear protein bands between 112 kDa–120 kDa for all recombinant PHO1 fused with the His label (Appendix A). Western blotting demonstrated that this protein band was specifically recognized by the PHO1-C antibody (Appendix A). In summary, these results indicate that PHO1-C protein was successfully expressed in *E. coli*.

#### 2.5.2. Enzyme Assay

The activity of PHO1 was determined by measuring the release of Pi from Glc-1-P in the presence of maltodextrins using purified recombinant mutated and non-mutated PHO1. All of the recombinant PHO1, both mutated and non-mutated, were found to have similar activity without any pre-treatment with ATP. However, to investigate the effect of phosphorylation site mutation on the activity of PHO1, recombinant mutated and non-mutated PHO1 were pre-treated with 1 mM ATP to stimulate protein phosphorylation by protein kinases present in the amyloplast lysate before enzyme assays. The results showed that S566 is a critical site for the upregulation of PHO1 activity, as PHO1S566A exhibited significantly lower activity compared to non-mutated PHO1 and PHO1S566E, presumably due to the lack of a phosphorylation site in PHO1S566A. The S69 mutation was also found to affect the activity of PHO1 but to a lesser extent. The results are presented in Figure 5.

## 3. Discussion

PHO1 is a major active form of phosphorylase in plants and has a diverse role across different species. In rice, PHO1 accounts for 96% of the total phosphorylase activity and is localized in the amyloplast matrix [32]. Similarly, PHO1 is the second most abundant protein in the amyloplast matrix of maize after SBEIIb [23], and its expression pattern in terms of transcript and protein levels has been found to be similar to those reported previously [23,32,33,34]. PHO1 has been suggested to be involved in maize starch synthesis and regulation during endosperm development [32,35,36]. PHO1 activity has been reported in *Zea mays* grains at all developmental stages and the lack of PHO1 activity in *Zea mays* shrunken 4 (Sh4) mutants resulted in reduced starch content [16,37]. Similarly, PHO1 expression patterns and activity have been reported in developing grains of *Hordeum vulgare* [38]. Expression analysis of other starch biosynthesis enzymes in *Zea mays* suggests that the expression patterns of SSs, SBEs, DBEs, and DPEs were similar to PHO1, and the activities of these enzymes appear similar to PHO1 as well [36]. In *Oryza sativa*, a mutant lack of PHO1 showed changes in seed size and starch content, highlighting the importance of PHO1 in starch biosynthesis [32]. Overall, our expression analysis supports the potential importance of PHO1 in starch biosynthesis or endosperm development.

PHO1 plays a crucial role in starch metabolism, particularly in non-photosynthetic cells, by regulating the activities of various enzymes through protein–protein interactions. It has been reported that most isoforms of DBEs, SSs, and DPEs form complexes with PHO1, but the exact fate of these interactions remains unclear. One proposed function of PHO1 in starch biosynthesis is that it acts on malto-oligosaccharides (MOs) to produce linear maltose of sufficient length for subsequent branching reactions by SBEs [39]. Moreover, PHO1 and SBEs isoforms have been shown to have functional interactions in the rice endosperm, indicating mutual capacities for chain elongation and branching [40]. PHO1 has also been suggested to be involved in the production of starch through functional or physical interactions with SSIV during starch biosynthesis [22,23,25,41]. Therefore, investigating the possible interactions of PHO1 with other starch biosynthetic enzymes is crucial for understanding the regulation and role of PHO1 in *Zea mays* amyloplast starch biosynthesis.

Phosphorylation is a key post-translational modification that regulates the activity and function of many enzymes. In starch biosynthesis, various enzymes including SBEI, SBEIIa, SBEIIb, and PHO1 have been reported to undergo phosphorylation [23,33]. The phosphorylation of serine residues in SBEIIa stimulates its activity, while dephosphorylation reduces it. Similarly, the phosphorylation of PHO1 is a critical feature during starch filling [42]. Specifically, S566 is a critical site for PHO1 activity, as mutations at this site significantly reduce its overall activity. Enzymatic analysis of purified recombinant PHO1 indicates that phosphorylation at S566 positively regulates PHO1 synthetic direction activity (the enzymatic activity is influenced by factors like temperature, pH, substrate and enzyme concentrations, inhibitors, activators, ionic strength, post-purification handling, and impurities, and therefore strict experimental controls are essential for consistency). The N-terminus of PHO1 may also play a role in regulating the L80 domain. The double mutant (mutated in both S69 and S566) can further illustrate the importance of both sites in their respective terminals. Overall, the results suggest that phosphorylation enhances the activity of PHO1 and regulates its ability to catalyze the reversible transfer of glucosyl units from Glc-1-P to the non-reducing end of α-1-4 linked glucan chains.

According to biochemical evidence, unlike the phosphorylation of PHO1, the available level of Pi may be responsible for determining its activity (phosphorolysis/synthesis) in starch metabolism [35,43]. The activity of PHO1 present in amyloplast may be affected by other enzymes such as SBE [40] and other metabolites such as Glc-1-P and Pi, which are considered a control mechanism for PHO1 activity in high Pi/G-1-P ratios [20]. However, the findings of Hwang suggest that Pi concentration in rice only partially affects the incorporation of [U14_C]-Glc-1-P into starch [42]. PHO1 in rice is capable of biosynthetic reactions even with physiological Glc-1-P substrate levels (0.2 mM) and a 50-fold excess of Pi in vitro compared to normal physiological levels. ADP-glucose, the main precursor for starch biosynthesis, inhibits the activity of PHO1 in the synthetic direction [17] and can reduce the activity of plastid PHO1 in amyloplast lysates. Although PHO1 has a lower affinity for Glc-1-P and is primarily thought to function in phosphorolysis rather than the synthesis of glucan polymers based on observations of glycogen phosphorylases in animal systems [35]. Recent findings suggest that PHO1 may indirectly affect the activity of other involved enzymes such as key starch-degrading enzymes (amylase) [44]. Even though PHO was discovered 75 years ago, its exact role is still being debated. Therefore, generating various mutant lines lacking PHO1 and comparing metabolites between mutants may help elucidate its major role in the starch biosynthesis pathway.

It is found that the stability and localization of PHO1 in *Zea mays* were not significantly affected by mutations in the phosphorylation sites, possibly due to the presence of other partners in the plastid. It is observed that the expression patterns of non-mutated PHO1 and two mutated forms were similar, indicating that phosphorylation at these sites may not play a significant role in localization. Interestingly, it was found that dephosphorylation of the S566 and S69 sites may enhance local expression patterns, suggesting a potential role for dephosphorylation in this process. Overall, the study suggests that the phosphorylation sites in PHO1 may not have a significant impact on stability and localization in *Zea mays*.

Previous studies have reported the phosphorylation of various PHO1 complexes [24], but no direct interaction between PHO1 and other starch biosynthesis enzymes has been reported. Attempts to discover a direct interaction between PHO1 and SBEs and SSs through yeast two-hybrid assays have not been successful so far. Co-immunoprecipitation analysis suggests that PHO1 can only form complexes with more than two enzymes. While the phosphorylation of an enzyme in a complex may influence its stability [24,38], phosphorylation of the complex may serve as a regulatory element that modulates enzymatic activity rather than determining the stability of the complex.

## 4. Conclusions

The transcript levels of PHO1 peak in a pattern that aligns closely with various starch biosynthetic enzymes. Analysis of ZmPHO1 expression suggests its involvement in the synthesis and regulation of maize starch during endosperm development. While the phosphorylation sites do not influence the stability and localization of PHO1, indicating that these sites do not play a role in these aspects for PHO1 in *Zea mays*, the phosphorylation of the S566 residue within the L80 region does regulate or amplify the activity of PHO1, emphasizing the significance of this specific phosphorylation site.

These results highlight the significant role of phosphorylation in modulating the activity of PHO1, a critical enzyme in the early and maturity stages of endosperm development. Therefore, any variations in its activity could have substantial consequences for starch production. Our study offers valuable insights that could enhance our comprehension of starch synthesis, with substantial implications for crop improvement. By generating and studying mutants with augmented or restricted PHO1 activity, our research opens avenues for the development of crop varieties with potentially higher starch content. This could lead to improvements in crop yield and quality. Furthermore, elucidating how the phosphorylation status of PHO1 influences PHO1 activity could pave the way for the creation of crop varieties with altered starch properties.

## 5. Materials and Methods

### 5.1. Preparation of Plant Material

An Mo17 inbred line of *Zea mays* seeds was selected to identify the expression level of PHO1 protein and transcripts. The inbred line was grown under normal field conditions at the Chongzhou Research Base of Sichuan Agricultural University, China. Materials were collected at 2-day intervals (from 2 to 30 DAP) and quick-frozen with liquid nitrogen and stored at –80 °C for later use.

#### 5.1.1. Transcripts of PHO1 Expression Pattern

The expression analysis in different developmental stages of seeds was carried out using the expression level data determined by RT-qPCR. The extraction of RNA, cDNA synthesis, and expression analysis was carried out as described by Jian M. [14]. The GAPDH gene encoding glyceraldehyde-phosphate dehydrogenase was used as the internal reference. The following primers were used to measure the transcript level of PHO1: Fwd: CTAACAGGACAATATGCA; Rev: GCTTCATTGGCCTTGGCA.

#### 5.1.2. PHO1 Protein Expression Pattern

The protein was extracted from the developing *Zea mays* seeds according to the protocol described in [45]. SDS-PAGE was performed using a Vertical Electrophoresis System (Bio-Rad, Shanghai, China). Proteins were separated in SDS-PAGE on 10% acrylamide gels. Anti-rabbit antibodies (ZmPHO1) were generated according to the protocol described in [28]. After electrophoresis, the proteins in polyacrylamide gels were transferred to nitrocellulose membranes using an Electrophoretic Transfer Cell (Bio-Rad). The transfer buffer contained 10% running buffer, 20% methanol, and 70% water. After this membrane was incubated in blocking buffer (5% solution of skim milk in 1% TBST) for one hour on a rotator and then incubated overnight at 4 °C. A 15 µL quantity of ZmPHO1 antibody (affinity pure antibody) was added with (Anti-PHO1) 1:1000 dilution in blocking buffer, then incubated for 1 h on rotator at room temperature. The gel membrane was washed three times for 10 min each in 1% TBST. HPR secondary binding antibody Rabbit IgG was added with 1:2000 dilution in blocking buffer. The gel membrane was incubated for one hour. The gel membrane was washed three times for 10 min each in 1% TBST. A resolving solution was added according to the company or manufacturer’s instruction to photograph the bands’ results.

#### 5.1.3. Phos-tag^TM^ Enrichment and Dephosphorylation

The experiment was performed as described by [28]. Phosphoproteins were selectively enriched using Phos-tag™ agarose as per the guidelines provided by the manufacturer, Wako Pure Chemical Industries Ltd., Osaka, Japan. In this process, we began with 200 µg of protein that was mixed with 200 µL of Zn^2+^-Phos-tag™ agarose. This mixture was subjected to an incubation process at a temperature of 4 °C for 4 h. Post incubation, the mixture was thoroughly washed thrice with the wash buffer composed of 0.1 M Tris-acetate, and 1 M sodium acetate with pH adjusted to 7.5. Next, an elution buffer containing 0.1 M Tris-acetate, 1 M sodium chloride, and 10 mM sodium phosphate (pH 7.5) was used to elute the enriched phosphoproteins. The elution process involved the application of an alkaline phosphatase reaction solution of 10 μL, which contained 0.5 mM magnesium chloride and 40 µg of protein with 0.002 units of alkaline phosphatase. A control reaction solution of 10 μL, including 0.5 mM magnesium chloride and 40 µg protein, was also prepared. Both solutions were subjected to a temperature of 37 °C for 3 h. Subsequently, Phos-tag™ was employed for further enrichment and elution of the proteins. Lastly, the enriched phosphoproteins were analyzed through SDS-PAGE and immunoblotting techniques to confirm the results.

#### 5.1.4. Phosphorylation Sites Prediction

The Expassy online tool (https://www.expassy.org/) was used (accessed on 5 December 2020) to predict the phosphorylation sites with default settings.

#### 5.1.5. iTRAQ^TM^ Labeling and Mass Spectrometry Analysis

The detection of phosphorylated proteins was accomplished through the utilization of the iTRAQ™ technique. This investigation was replicated three times, making use of protein samples from Mo17 maize 25 DAP kernels gathered both during the day and night. The process of sample lysis and protein extraction was executed using SDT buffer, comprising 4% SDS, 100 mM Tris-HCl, and 1 mM DTT at a pH of 7.6. The Bio-Rad BCA Protein Assay Kit (Hercules, CA, USA) was used to determine the protein concentration. Digestion of the protein was achieved with the help of trypsin, following the filter-aided sample preparation (FASP) procedure. The resulting peptides from each sample were subsequently desalted using Empore™ SPE Cartridges C18, standard density, then concentrated using vacuum centrifugation and reconstituted in a solution containing 0.1% (*v*/*v*) formic acid. The enrichment of phosphopeptide was conducted employing the High-Select™ Fe-NTA Phosphopeptides Enrichment Kit, as per the guidelines given by the manufacturer (Thermo Scientific, Waltham, MA, USA). Following lyophilization, the phosphopeptides were suspended once more in 20 µL of loading buffer consisting of 0.1% formic acid. The LC-MS analysis was executed with a timsTOF Pro mass spectrometer (Bruker, Beijing, China) in conjunction with Nanoelute (Bruker Daltonics) for 60 min. Peptides were placed onto a C18-reversed phase analytical column (25 cm length, 75 μm inner diameter, 1.9 μm, C18), initially with buffer A (0.1% formic acid), and then separated using a linear gradient of buffer B (84% acetonitrile and 0.1% formic acid) at a 300 g/min flow rate. The mass spectrometer was set to operate in positive ion mode, collecting ion mobility MS spectra within the *m*/*z* 100–1700 mass range and 1/k0 values varying from 0.6 to 1.6. This was followed by conducting 10 cycles of PASEF MS/MS, with a target intensity of 1.5 k and a threshold of 2500. The active exclusion feature was engaged with a release duration of 0.4 min. The initial data garnered from the mass spectrometry analysis were in the RAW file format, and subsequent library identification and quantitative analysis were conducted with the aid of the MaxQuant software (version 1.5.3.17).

### 5.2. PHO1 Gene Cloning and Recombination

Extracted RNA was used to generate cDNA, which was used to amplify the full-length *Zea mays* PHO1 (gene of PHO1), after the verification of sequence identity subjected to cloning. The following primers were used to amplify the full length of *Zea mays* PHO1: Fwd: ATGGCGACGACCACCT; Rev: GAGCACAGAAGAAAGAGCAAAG. The full-length sequence was initially cloned with T-vector (pMD19-T) using the kit from Takara, China, according to the abovementioned instructions and prepared PHO1-19t plasmid. PHO1-19t was used to amplify the full-length *Zea mays* PHO1 and after the verification of sequence identity was subjected to recombination with pET32a and pCAMBIA-2300-eGFP vectors. The following primers: Fwd: GCCATGGCTGATATCATGGCGACGACCACCTCG CCG; Rev: TTGTCGACGGAGCTCGAATTCGGGAAGGATGGC AGGGCTGAT, having an overhang sequence of pET32a, were used to recombine the full-length *Zea mays* PHO1 with pET32a vector to produce a PHO1-32a plasmid. The following primers: Fwd: ATTTGGAGAGGACAGGGTACCATGGCGACGACCACCTCG; Rev: GGTACTAGTGTCGACTCT AGACTAGGGAAGGATGGCAGGGC, having an overhang sequence of pCAMBIA-2300-eGFP, were used to recombine the full-length *Zea mays* PHO1 with pCAMBIA-2300-eGFP to produce a PHO1-2300 plasmid. *Eco*R I and *Eco*R V combination of restriction enzymes (bought from Takara, Dalian, China) was used for the linearization of the pET32a vector. The reaction mixture was prepared as follows: pET32a: 1 μg; *Eco*R V/*Eco*R 1: 2 μL; 10 × H Buffer: 2 μL; ddH_2_O: up to 20 μL. The combination of restriction enzymes *Kpn* I and *Xba* I (bought from Takara, China) was used for the linearization of the pCAMBIA-2300-eGFP vector. The reaction mixture was prepared as follows: pCAMBIA-2300-eGFP: 1 μg; *Kpn* I/*Xba* I: 2 μL; 10 × H Buffer: 2 μL; ddH_2_O: up to 20 μL. The reaction mixture was kept at 37 °C for up to 6 h for proper digestion. The product was subjected to running on 1% agarose gel and the linearized vector was purified using the Gel-DNA recovery mini kit (Takara, Dalian, China). The product was used directly or stored at 4 °C. A Clone-Express II One Step Cloning Kit (Takara, China) was used for recombination cloning. DH5α competent cells were used to transfer and amplify the respective plasmids. The clones that were grown were used for bacteria testing to ensure positive clones. The purified plasmid (purification was performed with a plasmid purification kit by Takara, China) was used for sequencing and gene confirmation and kept at 4 °C for further use.

### 5.3. Site-Directed Point Mutations

The purified plasmid PHO1-19t was used to mutate the phosphorylation sites serine 69 and serine 566 with base-pair residues which encode alanine, aspartic acid, and glutamic acid instead of serine at their respective position. QuickMutation™ Site-Directed Mutagenesis Kit by Beyotime was used for site-directed point mutations through a PCR reaction. The following primers were used to mutate the serine 69 residue site: Ala-Fwd: TGCAAGGCCCCGTCGCGCCCGCGGAAGGGCTT; Ala-Rev: AAGCCCTTCCGCGGG CGCGACGGGGCCTTGCA; Glu-Fwd: TGCAAGGCCCCGTCGAGCCCGCGGAAGGG CTT; Glu-Rev: AAGCCCTTCCGCGGGCTCGACGGGGCCTTGCA. The following primers were used to mutate the serine 566 residue site: Ala-Fwd: CAGAAGAAGAGAGTGCT GAGGATGAGTTAG; Ala-Rev: CTAACTCATCCTCAGCACTCTCTTCTTCTG; Glu-Fwd: AGAAGAAGAGAGTGAAGAGGATGAGTTAGATCC; Glu-Rev: GGATCTAACTCTTC CTCATCACTCTCTTCTTCT; Asp-Fwd: AGAAGAAGAGAGTGATGAGGATGAGTTAGATCC; Asp-Rev: GGATCTAACTCATCCTCATCACTCTCTTCTTCT. PCR was set up using the synthesized primers (from Takara, China) to amplify the complete plasmid following the Beyotime QuickMutation™ Site-Directed Mutagenesis Kit protocol. Following the PCR, the product was further incubated at 37 °C for 5 min after adding the 1 μL of *Dpn* I. The product was used directly or stored at 4 °C. DH5α competent cells were used to transfer and amplify the respective plasmids. The clones that were grown were used for bacteria testing to ensure positive clones. The purified plasmid (purification was performed by plasmid purification kit by Takara, China) was used for sequencing and gene confirmation and kept at 4 °C for further use.

### 5.4. Expression and Purification of Recombinant PHO1

BL21-receptive cells were used for the expression of protein induced via IPTG treatment. An His-tag protein purification kit bought from Beyotime, China, named “BeyoGold^TM^ His-tag Purification Resin” was used for the purification of the protein by following the steps in the instructions accompanying the product.

#### Enzyme Assay

The activity of PHO1 was determined using the method previously used by [16]. Recombinant PHO1 was used for the enzyme assays to demonstrate the mutation effect. ATP treatment was performed with the reaction mixture following the steps described by [27]. Briefly, the amyloplast extract from maize was mixed with bacterial lysate and 1 mM ATP to induce phosphorylation via protein kinase present in the amyloplast lysate. The reaction mixture was prepared in a final volume of 100 μL containing 1% maltodextrins, 75 mM glucose-l-P, 50 mM glycerophosphate buffer, pH 6.5, and enzyme extract followed by incubation at 30 °C for 5 or 10 min. The reaction was terminated by the addition of 2 mL of 0.072 N H_2_SO_4_. After this, 2 mL of 1 N H_2_SO_4_ acid solution was added containing 1% ammonium molybdate and 4% ferrous sulfate. After 2 min, the resultant color was read at 700 nm (Appendix A). The experiment was repeated 3 times.

### 5.5. Preparation of ZmPHO1 Antibody

The polyclonal antibody was generated as described in [46]. Subcutaneous immunization of New Zealand white rabbits (weighing 2 kg) was performed using 500 µg of the purified recombinant PHO1-C fusion protein. This protein was emulsified with 500 µL of Freund’s complete adjuvant in a 1:1 (*v*/*v*) ratio. Two weeks following the primary immunization, the rabbits received five subsequent subcutaneous injections every week. Each of these booster injections comprised 500 µg of the purified protein combined with 500 µL of Freund’s incomplete adjuvant, maintaining a 1:1 ratio. A week post the final injection, blood was drawn from the rabbit’s carotid artery, and serum was subsequently separated. The serum was treated with the addition of 15% glycerin and 0.2% sodium azide and then stored at a temperature of –80 °C for future use. The performance of the antibody serum was then assessed through a Western blot assay.

### 5.6. Maize Leaf Protoplast Transformation

Material treatment: *Zea mays* seeds were germinated in 1:1 of nutrient soil and zircon for about 3 days to about 1–2 cm in bud length, and transferred to a dark-environment culture at 25 °C for 10 days; the seedlings were exposed for a short time (<12 h) to low light (<378 mol*m^−2^s^−1^) before the extraction of the protoplast.

The extraction of protoplasts: the seed blades of the treatment were cut into 0.5–1 mm wide blades with razorblade, immersed in an enzymatic solution, placed in a light-absorbing vacuum for 30 min, kept for enzymatic reaction for 3.5 h followed by shaking by hand for 10 min, then the same volume of W5 solution was added to terminate the enzyme reaction, filtered with 75 µm nylon mesh, the filtrate fluid was centrifuged at 80 g for 5 min, precipitation was formed, the upper liquid was removed followed by the slow addition of 5 mL of W5 to re-suspend the pellet, this kept on ice for 30 min, the supernatant liquid was removed, MMG solution was added and the pellet re-suspended so that the final concentration of progenitor was 2 × 105/mL. Then, 20 μg recombinant particles were added to a 2 mL round bottom centrifuge tube bottom and 100 μL native mass was added, gently mixed, 100 µL PEG was added, the bottom of the tube mix was gently beaten, conversion was induced in dark conditions for 15 min, 400 µL W5 solution was added to terminate the reaction, 80× *g* centrifuged for 2 min, the upper clear liquid was removed, 100 μL WI solution was added, this was gently mixed at 23 °C with 2000 Lux light for 12 h to induce the protoplast, and a fluorescent microscope was used to observe the protoplast.

Enzymatic solution: 1.5% (*w*/*v*) Cellulase, 0.75% (*w*/*v*) Macerozyme, 20 mmol/L MES, 10 mmol/L KCl, 10 mmol/L CaCl_2_, 1% BSA, 0.6 mol/L mannitol. W5: 154 mmol/L NaCl, 125 mmol/L CaCl_2_, 5 mmol/L KCl, 2 mmol/L MES-KOH, pH = 5.7. MMG: 4 mmol/L MES, 0.4 mol/L Mannitol, 15 mmol/L MgCl_2_. PEG-Ca^2+^: 0.2 mol/L Mannitol, 0.1 mol/L CaCl_2_, (*w*/*v*) 25% PEG 4000. WI: 4 mmol/L MES, 0.5 mol/L Mannitol, 20 mmol/L KCl.

## Figures and Tables

**Figure 1 plants-12-03205-f001:**
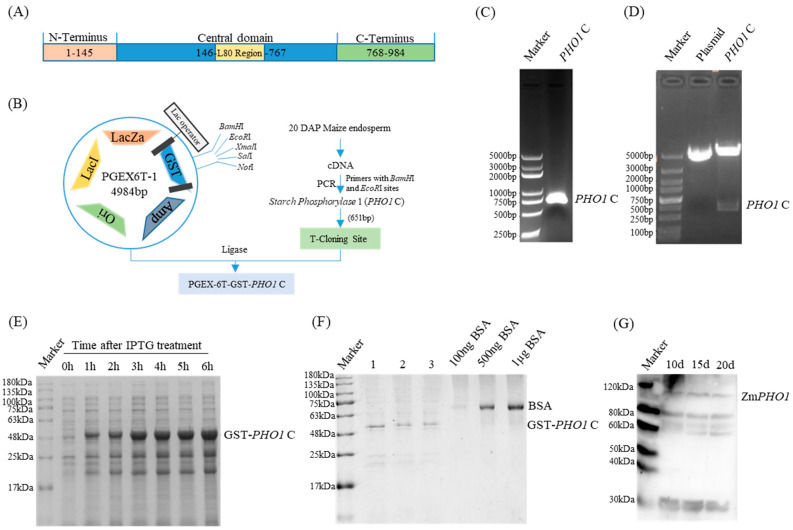
Induced expression of the recombinant plasmid and PHO1 rabbit antiserum-specific detection and verification of ZmPHO1 antibodies in maize tissue. (**A**) Symbolic representation of domains on the primary structure of *Zea mays* PHO1 (ZmPHO1); (**B**) recombinant plasmid vector construction flow chart. (**C**) PHO1-C-3′ end PCR amplification. (**D**) PGEX-6T-1-PHO1-C recombinant plasmid digestion identification. (**E**) IPTG induced expression of PHO1-C protein; (**F**) the targeted protein purified using GST-tag purification resin, with BSA as a loading control; (**G**) detection of specificity of antibody against ZmPHO1 of maize seed; the dilution ratio of the PHO1-C antibody was 1:1000.

**Figure 2 plants-12-03205-f002:**
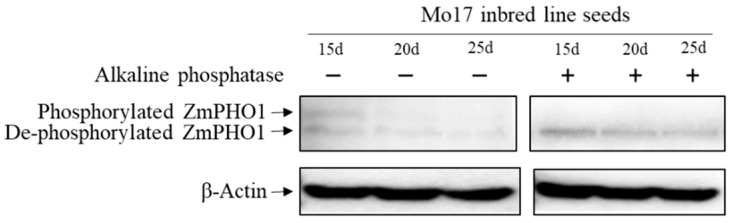
Western blot detection phosphorylation of *Zea mays* PHO1 (ZmPHO1). Negative and positive signs indicate the addition and absence of alkaline phosphatase. The dilution ratio of ZmPHO1 antibody was 1:1000 and the dilution ratio of β-Actin was 1:10,000. The amount of protein loaded was 30 µg.

**Figure 3 plants-12-03205-f003:**
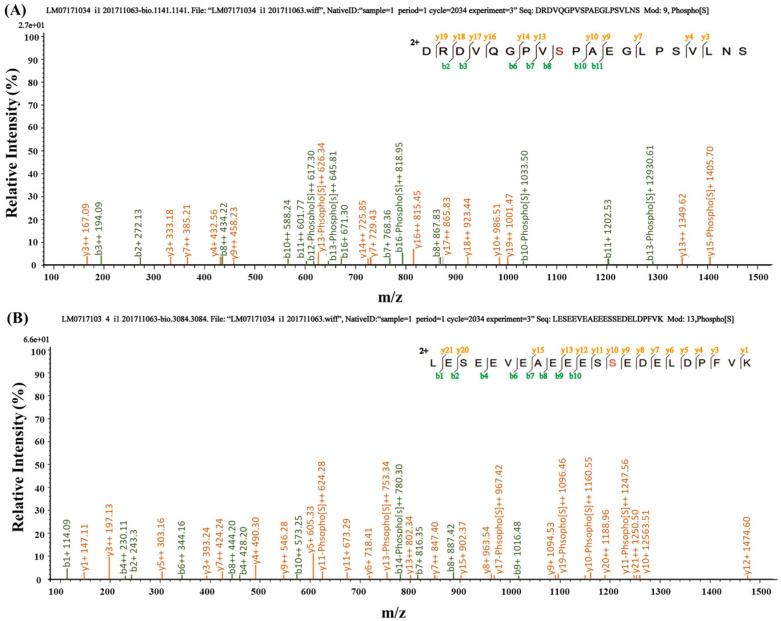
Mass spectrometry identification of ZmPHO1 phosphopeptide. (**A**) Mass spectrometry identification of ZmPHO1 phosphorylated peptide “DRDVQGPVSPAEGLPSVLNS”, the red letter (S) is the phosphorylation site at 69; (**B**) mass spectrometry identification of ZmPHO1 phosphorylated peptide “LESEEVEAEEESSEDELDPFVK”, the red letter (S) is the phosphorylation site at 566.

**Figure 4 plants-12-03205-f004:**
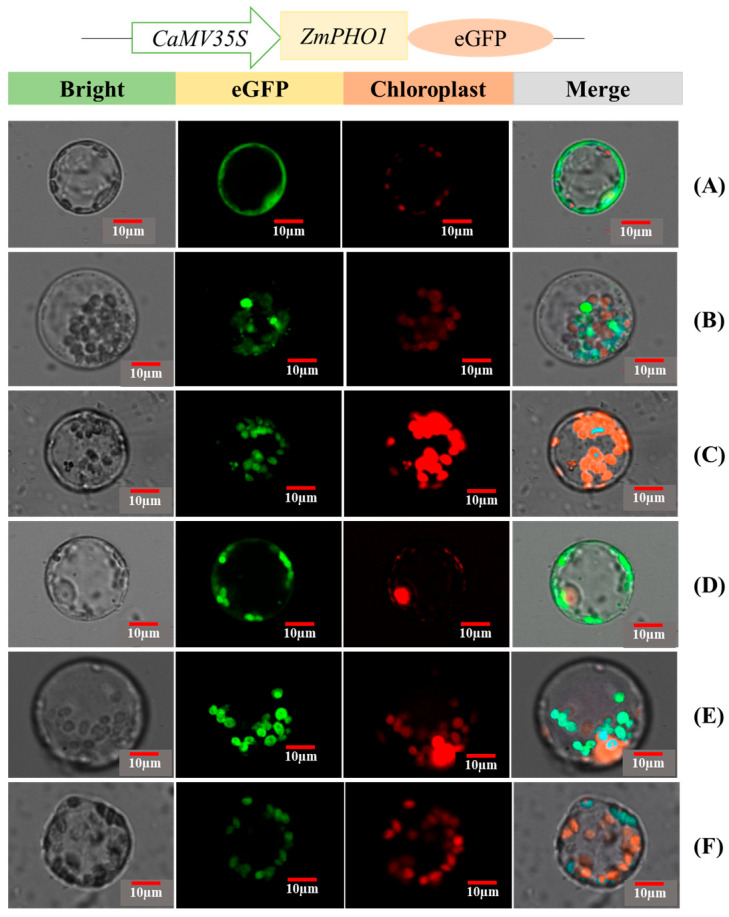
Expression and subcellular localization of mutated and non-mutated recombinant PHO1 in *Zea mays* protoplast of the leaves. (**A**) pCAMBIA2300-eGFP. (**B**) pCAMBIA2300-ZmPHO1-eGFP. (**C**) pCAMBIA2300-ZmPHO1-S566A-eGFP. (**D**) pCAMBIA2300-ZmPHO1-S566E-eGFP. (**E**) pCAMBIA2300-ZmPHO1-S69A-eGFP. (**F**) pCAMBIA2300-ZmPHO1-S69E-eGFP. All of the expressed GFP-tagged PHO1 is found in the chloroplast. There are no significant differences in the subcellular localization found for both phosphorylation site mutations.

**Figure 5 plants-12-03205-f005:**
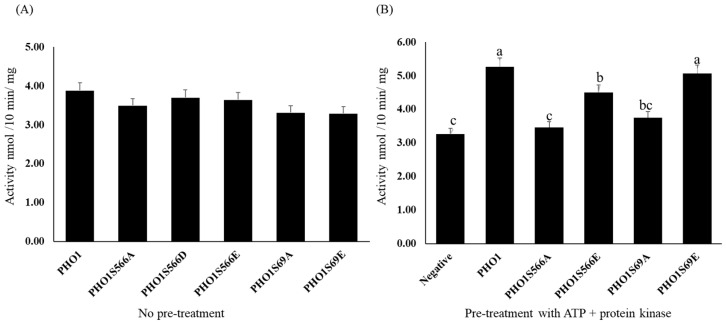
The activity of recombinant PHO1. (**A**) The activity of recombinant PHO1 with maltodextrin substrate with no pre-treatment with ATP. (**B**) The activity of phosphorylated PHO1 with maltodextrin substrate. Non-mutated recombinant PHO1 was used as a negative control without pretreatment with ATP. Activities were calculated as nmol/10 min/mg. Significantly different means (at *p* < 0.05) from the one-way ANOVA followed by LSD. a–c represents a significant difference.

**Table 1 plants-12-03205-t001:** iTRAQ identification of ZmPHO1 phosphorylation sites in N-terminal and L80 domain.

Credibility	Peptide Sequence	Phosphorylation Site	ZmPHO1 Site
99.35000038	DRDVQGPVSPAEGLPSVLNS	Serine at number 9	Serine 69
99.00000095	LESEEVEAEEESSEDELDPFVK	Serine at number 13	Serine 566

Note: This provides a basis for the study of maize PHO1 phosphorylation. The two sites Serine 69 and Serine 566 of PHO1 may be phosphorylated in maize.

## Data Availability

Not applicable.

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
