# Peer review of "Site-Directed Mutations at Phosphorylation Sites in Zea mays PHO1 Reveal Modulation of Enzymatic Activity by Phosphorylation at S566 in the L80 Region"

_plants, 2023, doi:10.3390/plants12183205_

Round 1

Reviewer 1 Report (New Reviewer)

The paper by  Shoaib et al clearly presents a methodology that reveals the functional implication of the S566 site in the L80 region of PHO1 of Zea mays in modulating its enzymatic activity by phosphorylation of the site. they employ several techniques such as biochemical (antibody preparation and specificity detection), bioinformatics, PCR and microscopy. Their methods and approaches are clearly described and justify the results. I would only suggest to make materials and methods description less detailed e.g. there is no need for the 4.6 section.

Author Response

The paper by  Shoaib et al clearly presents a methodology that reveals the functional implication of the S566 site in the L80 region of PHO1 of Zea mays in modulating its enzymatic activity by phosphorylation of the site. they employ several techniques such as biochemical (antibody preparation and specificity detection), bioinformatics, PCR and microscopy. Their methods and approaches are clearly described and justify the results. I would only suggest to make materials and methods description less detailed e.g. there is no need for the 4.6 section. Author response: We are grateful to the reviewer for their careful reading of our paper and their insightful suggestions. We have considered feedback and made the necessary adjustments accordingly. The mentioned section has been omitted and further adjustments are being made. For ease of identification, all changes have been marked in red in the manuscript file. We would like to extend our heartfelt gratitude once again for your meticulous review and insightful suggestions. Your point-to-point reading and careful observations have been instrumental in enhancing the quality of our manuscript. Such thorough feedback reflects your dedication to the advancement of scientific literature, and we deeply appreciate the time and effort you have invested We look forward to hearing from you regarding our submission. We would be glad to respond to any further questions and comments that you may have.

Thanks!

Reviewer 2 Report (New Reviewer)

I suggest the authors move Fig 1, 4, and Fig 6a and 6b to supplementary. Instead, they should include a structural figure of PHO1 showing the relevant residues (S69 and S566) in relation to substrates and/or activators/inhibitors. A relevant discussion should be included in the discussion part as well.  

Author Response

I suggest the authors move Fig 1, 4, and Fig 6a and 6b to supplementary. Instead, they should include a structural figure of PHO1 showing the relevant residues (S69 and S566) in relation to substrates and/or activators/inhibitors. A relevant discussion should be included in the discussion part as well.  

Author response: We are grateful to the reviewer for their careful reading of our paper and their insightful suggestions. We have considered feedback and made the necessary adjustments accordingly. We have shifted the figure 4, 6a and 6b to the supplementary section. For ease of identification, all changes have been marked in red in the manuscript file. The structural figure mentioning phosphorylation sites are included in the supplementary data. Moreover, we have included a conclusion section to summarize the manuscript prospect.  

Figure S4. Representation and conservation of identified modifiable (phosphorylation) serine residues of PHO1. (A) Symbolic representation of phosphorylation sites on the primary structure of Zea mays PHO1 (ZmPHO1). Red letters are representing modifiable residue identified via iTRAQTM.  (B) Conservation of Serine 69 (from Zea mays) phosphorylation site. (C) Conservation of Serine 566 (from Zea mays) phosphorylation site. Symbols include: Zm, Zea mays (accession id: NP_001296783.1); Sb, Sorghum bicolor (accession id: XP_021306483.1); Si, Setaria italica (accession id: XP_004981704.1); Ph, Panicum hallii (accession id: XP_025795811.1); Bd, Brachypodium distachyon (accession id: XP_003559211.1); Ta, Triticum aestivum (accession id: ACC59201.1); Hv, Hordeum vulgare (accession id: KAE8783983.1); Os, Oryza sativa (accession id: XP_015631420.1); At, Arabidopsis thaliana (accession id: Q9LIB2.1); Sl, Solanum lycopersicum (accession id: NP_001362574.1); St, Solanum tuberosum (accession id: NP_001275215.1); and Ca, Capsicum annuum (accession id: XP_016569840.1).

We would like to extend our heartfelt gratitude once again for your meticulous review and insightful suggestions. Your point-to-point reading and careful observations have been instrumental in enhancing the quality of our manuscript. Such thorough feedback reflects your dedication to the advancement of scientific literature, and we deeply appreciate the time and effort you have invested

We look forward to hearing from you regarding our submission. We would be glad to respond to any further questions and comments that you may have.

Thanks!

Reviewer 3 Report (New Reviewer)

The contribution plants-2536168 investigated how site-directed Mutations at phosphorylation sites in Zea mays PHO1 reveal modulation of enzymatic activity by phosphorylation at S566 in the L80 region. The manuscript shows a relevant topic. Overall, it is a good paper with a logical writing strategy.

There are some minor points:

Introduction section. A representative schema showing the starch phosphorylase (PHO) action mechanism would be helpful.

Why was Mo17 inbred line of Zea mays seeds selected to identify the expression level of  PHO1 protein and transcripts? Please explain.

Was the activity of PHO1 reported based on mg of protein content? If so, how was the protein quantified?

Figure 3, mass spectrometry identification of ZmPHO1 phosphopeptide must be improved; green and red text cannot be read.

In Figure 5, the scale bars can not be seen; please improve.

Regarding PHO1 activity, many factors can influence enzymatic activity, such as substrate concentration, pH, enzyme affinity, etc. The author needs to discuss these factors.

A Conclusion paragraph including some perspectives is necessary.

Author Response

The contribution plants-2536168 investigated how site-directed Mutations at phosphorylation sites in Zea mays PHO1 reveal modulation of enzymatic activity by phosphorylation at S566 in the L80 region. The manuscript shows a relevant topic. Overall, it is a good paper with a logical writing strategy.

Author response: We are grateful to the reviewer for their careful reading of our paper and their insightful comments and suggestions. We have considered all feedback and made the necessary adjustments accordingly. For ease of identification, all changes have been marked in red in the manuscript file.

There are some minor points:

Introduction section. A representative schema showing the starch phosphorylase (PHO) action mechanism would be helpful.

Author response: Thank you for your valuable feedback. In our previous work, we provided a detailed schematic representation which led us to believe a textual elaboration with references would suffice. However, based on your recommendation, we have enhanced the section detailing PHO1's role in starch metabolism. We hope this revised description offers greater clarity and addresses any lingering ambiguities.

Following changes are incorporated in the Introduction section (Lines 52-68).

The expression patterns and subcellular localization of PHO1, particularly within plastids underscore its significant role in starch biosynthesis, as evidenced by correlations with several starch biosynthetic enzymes [8,14]. Starch biosynthesis is a multifaceted process engaging over 15 enzymes, including starch synthases (SSs, EC 2.4.1.21), starch branching enzymes (SBEs, EC 2.4.1.18), and starch de-branching enzymes (DBEs, 3.2. 1.41; predominantly isoamylases (ISAs)). Each enzyme has specific tissue- and development-specific isoforms [15]. PHO1 may play an indispensable role during the early stages of endosperm development, as demonstrated by its higher activity than SSs during initial maize kernel development stages [16]. The biochemical evidence suggest that PHO1 acts on glucans to reversibly produce Glc-1-P and Pi [11]. In the synthetic pathway, PHO1 facilitates the transfer of glucosyl units from Glc-1-P to expanding glucan chains, releasing Pi. In contrast, the phosphorolytic route sees the same reaction in the reverse direction [17-19]. The activity of PHO1 in starch biosynthesis is dictated by the Pi/Glc-1-P ratio, with higher ratios favoring glucan synthesis, and lower ratios favoring degradation [20]. However, factors beyond substrate availability can also steer the direction of PHO1 activity [21]. This understanding adds an essential layer of complexity to the regulation of starch biosynthesis, warranting further exploration into the factors controlling PHO1 activity.

Why was Mo17 inbred line of Zea mays seeds selected to identify the expression level of  PHO1 protein and transcripts? Please explain.

Author response: Some inbred lines, like B73 and Mo17, have been historically used in maize genetics and breeding studies. Their genetic background is well-characterized, which provides a consistent platform for comparative research. Mo17 represents a different genetic background compared to other commonly studied maize lines. Using diverse genetic backgrounds can provide insights into how genes are expressed across different maize genotypes. Moreover, multiple labs in our university are working on same lines, using the same inbred line can help ensure consistency and comparability across studies.

Was the activity of PHO1 reported based on mg of protein content? If so, how was the protein quantified?

Author response: When the activity of PHO1 in Zea mays was assessed, the results were standardized based on protein content, providing a consistent metric for enzyme activity across various samples or treatments. This normalization is important, ensuring that differences in enzymatic activity are attributable to genuine variations in the enzyme performance and not merely due to inconsistencies in protein concentration. The Bradford assay was chosen for protein quantification because of its simplicity, quickness, and specificity to proteins. In this method, the Coomassie Brilliant Blue G-250 dye binds to proteins, resulting in a color shift, which is then quantified spectrophotometrically. The intensity of the blue color, measured typically at 595 nm, correlates directly with the protein concentration in the sample. The reported activity metric "nmol/10 min/mg" gives a clear insight into the enzyme's performance. It signifies the amount of substrate (in nmol) that the enzyme converts in 10 minutes per milligram of protein. 

Figure 3, mass spectrometry identification of ZmPHO1 phosphopeptide must be improved; green and red text cannot be read.

Author response: Thank you for your valuable feedback. The image quality has greatly been improved and is clearly presenting phosphorylation sites in specified peptide.

Following changes have been made (Figure 3)

Figure 3. Mass spectrometry identification of ZmPHO1 phosphopeptide. (A) Mass spectrometry identification of ZmPHO1 phosphorylated peptide “DRDVQGPVSPAEGLPSVLNS”, the red letter (S) is the phosphorylation site at 69; (B) mass spectrometry identification of ZmPHO1 phosphorylated peptide “LESEEVEAEEESSEDELDPFVK”, the red letter (S) is the phosphorylation site at 566.

In Figure 5, the scale bars can not be seen; please improve.

Author response: Thank you underline this deficiency. The scale bars are improved and incorporated in the figure.

Following changes have been made (Lines 236-242).

Figure 4. Expression and subcellular localization of mutated and non-mutated recombinant PHO1 in Zea mays protoplast of the leaves. (A) pCAMBIA2300-eGFP. (B) pCAMBIA2300-ZmPHO1-eGFP. (C) pCAMBIA2300-ZmPHO1-S566A-eGFP. (D) pCAMBIA2300-ZmPHO1-S566E-eGFP. (E) pCAMBIA2300-ZmPHO1-S69A-eGFP. (D) pCAMBIA2300-ZmPHO1-S69E-eGFP. All of the expressed GFP-tagged PHO1 is found in the chloroplast. There are no significant differences in the subcellular localization found for both phosphorylation site mutations.

Regarding PHO1 activity, many factors can influence enzymatic activity, such as substrate concentration, pH, enzyme affinity, etc. The author needs to discuss these factors.

Author response: We appreciate the reviewer's astute observation. Indeed, numerous factors can impact enzymatic activity. To mitigate variations stemming from substrate and enzyme concentration, we ensured that an equal amount of protein was loaded with a specified substrate concentration. Nonetheless, there remain inherent uncertainties that could potentially influence the enzyme's activity. In light of this feedback, we have added a section detailing potential errors and considerations that may affect enzymatic activity, ensuring a more comprehensive understanding for our readers. We have added a Note text in the figure legend.

Following additions have been made (Figure 5 legend).

Figure 5. The activity of recombinant PHO1. (A) The activity of recombinant PHO1 with maltodextrin substrate with no pre-treatment with ATP. (B) The activity of phosphorylated PHO1 with maltodextrin substrate. Non-mutated recombinant PHO1 was used as a negative control without pretreatment with ATP. Activities were calculated as nmol/ 10 min/ mg. Significantly different means (at P<0.05) from the One-way ANOVA followed by LSD. Note: Enzymatic activity is subject to a range of influencing factors. Variations in temperature or pH can significantly modulate performance, with each enzyme having its own optimal conditions. Moreover, the concentrations of the substrate and enzyme, as well as the presence of external molecules like inhibitors or activators, play critical roles. The ionic strength of the medium, post-purification handling, and even sample impurities can further affect outcomes. Hence, strict experimental controls are essential to ensure consistent and accurate results.

A Conclusion paragraph including some perspectives is necessary.

Author response: Thank you to identify the lacking passage. We have included the conclusion section.

Following changes have been made (Section: 4 Conclusion)

  1. Conclusion

The transcript levels of PHO1 peak in a pattern that aligns closely with various starch biosynthetic enzymes. Analysis of ZmPHO1 expression suggests its involvement in the synthesis and regulation of maize starch during endosperm development. While the phosphorylation sites do not influence the stability and localization of PHO1, indicating that these sites do not play a role in these aspects for PHO1 in Zea mays, the phosphorylation on the S566 residue within the L80 region does regulate or amplify the activity of PHO1, emphasizing the significance of this specific phosphorylation site.

These results highlight the significant role of phosphorylation in modulating the activity of PHO1, a critical enzyme in the early and maturity stages of endosperm development. Therefore, any variations in its activity could have substantial consequences on starch production. Our study offers valuable insights that could enhance our comprehension of starch synthesis, with substantial implications for crop improvement. By generating and studying mutants with augmented or restricted PHO1 activity, our research opens avenues for the development of crop varieties with potentially higher starch content. This could lead to improvements in crop yield and quality. Furthermore, elucidating how the phosphorylation status of PHO1 influences PHO1 activity could pave the way for the creation of crop varieties with altered starch properties.

We would like to extend our heartfelt gratitude once again for your meticulous review and insightful suggestions. Your point-to-point reading and careful observations have been instrumental in enhancing the quality of our manuscript. Such thorough feedback reflects your dedication to the advancement of scientific literature, and we deeply appreciate the time and effort you have invested

We look forward to hearing from you regarding our submission. We would be glad to respond to any further questions and comments that you may have.

Thanks!

Reviewer 4 Report (New Reviewer)

The present article: „Site-directed Mutations at Phosphorylation Sites in Zea mays PHO1 Reveal Modulation of Enzymatic Activity by Phosphorylation at S566 in the L80 Region“ deals with an important topic, but some passages are confusing and will need to be better explained.

line 17 abbreviation GT (as glycosyltransferases ?) is not explained

line 61 is confusing. In this article PHO1 is presented as starch synthesis enzyme (which is possible), however phosphorolysis should not be confused with starch synthesis. It's a term for the direction of starch degradation.

“PHO1 catalyzing glucan phosphorolysis to generate Glc-1-P and Pi”.

line 66 is confusing because phosphorolysis and degradation both degrade the starch. Didn´t you rather mean favouring phosphorolysis, and lower ratios favour glucan synthesis?

line 75 Which model crops have been studied so far?

line 76, please explain AGPase as ADP-glucose pyrophosphorylase, it has not been introduced yet.

line 91 A broader introduction to this section will be needed than starting with a shortcut pMD-19t-PHO1. In addition, neither Fig. 1C nor Fig. 1A,B provide any further information on pMD-19t-PHO1 and readers need not be familiar with the P-GEX-6T vector and GST.

line 100 to 111 The discrepancy between the theoretical molecular weight of 45 kDa and the 112 kDa found in Zea mays should be better explained

Fig. 1 and others, A,B,C,…are usually placed above the graph.

line 173 Please explain the iTRAQ abbreviation.

line 174-177 Paragraph rather belong to the discussion section.

Fig. 3 The labels in the chart are illegible

In section 2.2.6 it is not clear what is the result of this work and what is the discussion. References should preferably not be part of the Results.

line 258, 259 kDa is more often than KDa and is used previously

line 262 Can't the enzyme activity be affected by the procedures performed, especially maize leaves protoplast transformation rather than phosphorylation? Purification of recombinant PHO1 and non-mutated enzyme is very different. It is not discussed.

line 283. It is a specific activity related to total protein content?

The discussion is very good, but a clear conclusion would be helpful. The role of phoshorylation in PHO1 is still not obvious.

References should be presented in a consistent style. More than one author and et al would be needed. In addition, reference style manager misstates the surname in the abbreviation: e.g. ref. 1 Cori, G.T.C.F.C.S., G

line 647 Plos One

line 596 Int J Mol Sci instead of International Journal of Molecular Sciences etc..

I recommend the article for publication after all these minor revisions.

Minor editing of English language required

Author Response

The present article: „Site-directed Mutations at Phosphorylation Sites in Zea mays PHO1 Reveal Modulation of Enzymatic Activity by Phosphorylation at S566 in the L80 Region“ deals with an important topic, but some passages are confusing and will need to be better explained.

Author response: We are grateful to the reviewer for their careful reading of our paper and their insightful comments and suggestions. We have considered all feedback and made the necessary adjustments accordingly. For ease of identification, all changes have been marked in red in the manuscript file.

line 17 abbreviation GT (as glycosyltransferases ?) is not explained

Author response: Sorry, we forgot to add the abbreviation. The abbreviation has been added.

Following changes have been made (Line 17-18).

Starch phosphorylase (PHO) is a pivotal enzyme within the GT35-glycogen –phosphorylase (GT; glycosyltransferases) superfamily.

line 61 is confusing. In this article PHO1 is presented as starch synthesis enzyme (which is possible), however phosphorolysis should not be confused with starch synthesis. It's a term for the direction of starch degradation.

“PHO1 catalyzing glucan phosphorolysis to generate Glc-1-P and Pi”.

Author response: Sorry, we failed to express thoughts clearly. We have modified the text.

Following changes have been made (Line 61-63).

The biochemical evidence suggest that PHO1 acts on glucans to reversibly produce Glc-1-P and Pi [11]. In the synthetic pathway, PHO1 facilitates the transfer of glucosyl units from Glc-1-P to expanding glucan chains, releasing Pi.

line 66 is confusing because phosphorolysis and degradation both degrade the starch. Didn´t you rather mean favouring phosphorolysis, and lower ratios favour glucan synthesis?

Author response: Sorry that was a mistake, we failed to incorporate the text properly. The text have been updated.

Following changes have been made (Line 65-66).

The activity of PHO1 in starch biosynthesis is dictated by the Pi/Glc-1-P ratio, with higher ratios favoring glucan synthesis, and lower ratios favoring degradation.

line 75 Which model crops have been studied so far?

Author response: Thank you for underlining this deficiency. We forgot to mention the details which we have specified now.

Following changes have been made (Line 72-75).

Similarly, PHO1 phosphorylation has been identified as a crucial element during starch filling with its regulation via protein phosphorylation observed across various model crops including Zea mays, Oryza sativa, Hordeum vulgare, and Triticum aestivum [23-27].

line 76, please explain AGPase as ADP-glucose pyrophosphorylase, it has not been introduced yet.

Author response: Thank you do identify the mistake. The explanation has been added.

Following changes have been made (Line 75-76).

Moreover, phosphorylation of the brittle 2 (Bt2) subunit of ADP-glucose pyrophosphorylase (AGPase) was found to boost its activity in Zea mays [28].

line 91 A broader introduction to this section will be needed than starting with a shortcut pMD-19t-PHO1. In addition, neither Fig. 1C nor Fig. 1A,B provide any further information on pMD-19t-PHO1 and readers need not be familiar with the P-GEX-6T vector and GST.

Author response: Sorry there were some mistakes in the texts. We have updated the section accordingly.

Following changes have been made (Line 92-104).

For the protein expression and purification, the PGEX vector system was employed. The PGEX-6T vector allows for the expression of fusion proteins tagged with Glutathione S-transferase (GST), facilitating easy purification. GST is a widely used tag due to its robust expression and ease of purification using glutathione-based resins. The 651 bp fragment from the C-terminal of PHO1 (PHO1-C) gene was amplified (Figure 1C) and cloned to pGEX-6T vector to generate PGEX-6T-PHO1-C plasmid. The PGEX-6T-PHO1-C expression vector was successfully constructed by double-digesting the recombinant plasmid PGEX-6T-PHO1-C with BamHI and EcoRI, obtaining a fragment with a band size of 651 bp, which was consistent with the expected results (Figure 1D). Further sequencing of the positive clones of the recombinant plasmid confirmed the successful construction of the PGEX-6T-PHO1-C expression vector.

line 100 to 111 The discrepancy between the theoretical molecular weight of 45 kDa and the 112 kDa found in Zea mays should be better explained

Author response: Thank you for the query. The theoretical molecular weight for our protein of interest was deduced to be 45 kDa, as determined from its corresponding amino acid sequence derived from a 651 bp segment (utilized for antibody preparation) of the entire 2955 bp gene. This was computed by adjusting for the weight of water lost during peptide bond formation using the ExPASy ProtParam tool. Notably, the full gene translates to a protein of approximately 112 kDa. We have described it more clearly in the text now.

Following changes have been made (Lines 91-117).

2.1. Antibody Preparation and Specificity Detection

For the protein expression and purification, the PGEX vector system was employed. The PGEX-6T vector allows for the expression of fusion proteins tagged with Glutathione S-transferase (GST), facilitating easy purification. GST is a widely used tag due to its robust expression and ease of purification using glutathione-based resins. The 651 bp fragment from the C-terminal of PHO1 (PHO1-C) gene was amplified (Figure 1C) and cloned to pGEX-6T vector to generate PGEX-6T-PHO1-C plasmid. The PGEX-6T-PHO1-C expression vector was successfully constructed by double-digesting the recombinant plasmid PGEX-6T-PHO1-C with BamHI and EcoRI, obtaining a fragment with a band size of 651 bp, which was consistent with the expected results (Figure 1D). Further sequencing of the positive clones of the recombinant plasmid confirmed the successful construction of the PGEX-6T-PHO1-C expression vector.

The constructed prokaryotic expression vector PGEX-6T-PHO1-C was transformed into the E. coli BL21 (DE3) strain, which was then induced with 0.5 mmol/L IPTG. After being identified via 10% SDS-PAGE, a significant number of recombinant PHO1 protein bands were obtained from 48 kDa–55 kDa, consistent with the expected theoretical molecular weight (about 48 kDa; calculated based on its amino acid sequence, adjusting for the weight of water released during peptide bond formation, using the ExPASy ProtParam tool) (Figure 1E). The concentration of the recombinant protein was measured and compared to that of the BSA control, which showed that the three concentrations of the recombinant protein GST-PHO1-C were significantly higher than those of the BSA control. These results indicated that the GST-PHO1-C recombinant protein was successfully purified and could be used as an antigen for the preparation of an antibody. To investigate whether the PHO1 antibody specificity, total protein was extracted from maize seeds in several developmental stages, and bands of similar size were detected using Western blotting at approximately 112 kDa (Figure 1G), consistent with the predicted PHO1 size in maize.

Fig. 1 and others, A,B,C,…are usually placed above the graph.

Author response: Thank you for underlining this deficiency. The figures in the complete manuscript are being updated, and A B C is placed above the figure.

line 173 Please explain the iTRAQ abbreviation.

Author response: Thank you to underline this. We have added the abbreviation.

Following changes have been made (Line 179).

2.2.5. iTRAQ (isobaric tags for relative and absolute quantitation) Identification of ZmPHO1 Phosphorylation Sites

line 174-177 Paragraph rather belong to the discussion section.

Author response: Thank you to underline this. Previously we failed to express the thoughts appropriately which we have updated in the revised version.

Following changes have been made (Line 181-187).

We utilized Phos-tagTM technology to enrich phosphorylated proteins from endosperm samples collected at different stages of maize pollination and identified a large number of phosphorylated proteins involved in the starch synthesis pathway through mass spectrometry analysis. Based on these findings, we speculated that ZmPHO1 may be phosphorylated by kinases. Further analysis using the iTRAQ revealed two specific phosphorylation sites, namely Serine 69 in the N-terminal peptide and Serine 566 in the L80 domain of PHO1 (Table 1, Figure 3).

Fig. 3 The labels in the chart are illegible

Author response: Thank you for underlining this. We have updated the figure 3 and hoping it is more clear.

Following changes have been made (Figure 3).

Figure 3. Mass spectrometry identification of ZmPHO1 phosphopeptide. (A) Mass spectrometry identification of ZmPHO1 phosphorylated peptide “DRDVQGPVSPAEGLPSVLNS”, the red letter (S) is the phosphorylation site at 69; (B) mass spectrometry identification of ZmPHO1 phosphorylated peptide “LESEEVEAEEESSEDELDPFVK”, the red letter (S) is the phosphorylation site at 566.

In section 2.2.6 it is not clear what is the result of this work and what is the discussion. References should preferably not be part of the Results.

Author response: Sorry it was a mistake. We have removed the section and updated in the discussion.

Following changes have been made (Section 2.2.6).

2.2.6. Conservation of identified phosphorylation sites in monocots and dicots

Multiple alignments were performed to determine the conservation of S69 and S566 phosphorylation sites in various model crops from monocots and dicots. The results showed that the serine in both domains was highly conserved only in monocot crops. While dicot crops had aspartic acid and glutamic acid instead of serine in the N-terminus, no serine, aspartic acid, and glutamic acid was present on S566.

line 258, 259 kDa is more often than KDa and is used previously

Author response: Sorry it was a mistake. The section is aligning with the expression as written in whole manuscript (kDa).

Following changes have been made (Line 249-254).

The molecular weight of the ZmPHO1 protein fused with His label was around 120kDa. SDS-PAGE electrophoresis revealed clear protein bands between 112kDa-120kDa for all recombinant PHO1 fused with His label (Figure S8, S9). Western blotting demonstrated that this protein band was specifically recognized by PHO1-C antibody (Figure S8, S9). In summary, these results indicate that PHO1-C protein was successfully expressed in E. coli.

line 262 Can't the enzyme activity be affected by the procedures performed, especially maize leaves protoplast transformation rather than phosphorylation? Purification of recombinant PHO1 and non-mutated enzyme is very different. It is not discussed.

Author response: Indeed, the reviewer brings up a crucial point. The processes involved in maize leaves protoplast transformation and the purification methods for recombinant and non-mutated enzymes might introduce variables that can influence enzyme activity. These variables could range from changes in protein conformation to potential post-translational modifications or interactions with other cellular components. While our primary focus was on the phosphorylation sites and their direct impact on PHO1 activity, it is essential to acknowledge that the experimental procedures could also play a role. In future studies, we aim to optimize and standardize the purification methods for both recombinant and non-mutated enzymes to minimize any discrepancies. Furthermore, the differential behaviour observed during maize protoplast transformation, as well as any potential effects on enzyme functionality, will be a subject of in-depth investigation. We appreciate the reviewer's astute observation, and we elaborated on these aspects in the revised manuscript. The protocol followed was same if the reviewer asking for difference in band appearance, then there could be several reasons.

The introduction of mutations can sometimes lead to conformational changes in the protein structure. This might result in differences in migration patterns on a gel, even if the proteins have the same molecular weight.

Mutations can influence post-translational modifications (PTMs). For instance, the mutated site might be a phosphorylation or glycosylation site, and its alteration could result in a change in the overall mass or charge of the protein.

The purification process often relies on binding affinity, and mutations can influence the protein's binding characteristics. For example, if the mutation affects the protein's ability to bind to a column or a specific ligand used in the purification process, this could lead to variances in purification efficiency.

Some mutations can result in the protein aggregating or forming multimers, which would migrate differently on a gel compared to the monomeric form.

line 283. It is a specific activity related to total protein content?

Author response: Thank you for the query. Yes, it is specific activity for PHO1 from the total protein content calculated in different developmental stages.  

The discussion is very good, but a clear conclusion would be helpful. The role of phoshorylation in PHO1 is still not obvious.

Author response: Thank you for the suggestion. We have incorporated the conclusion section in the modified version of the manuscript.

Following changes have been made (Section 4: Conclusion)

  1. Conclusion

The transcript levels of PHO1 peak in a pattern that aligns closely with various starch biosynthetic enzymes. Analysis of ZmPHO1 expression suggests its involvement in the synthesis and regulation of maize starch during endosperm development. While the phosphorylation sites do not influence the stability and localization of PHO1, indicating that these sites do not play a role in these aspects for PHO1 in Zea mays, the phosphorylation on the S566 residue within the L80 region does regulate or amplify the activity of PHO1, emphasizing the significance of this specific phosphorylation site.

These results highlight the significant role of phosphorylation in modulating the activity of PHO1, a critical enzyme in the early and maturity stages of endosperm development. Therefore, any variations in its activity could have substantial consequences on starch production. Our study offers valuable insights that could enhance our comprehension of starch synthesis, with substantial implications for crop improvement. By generating and studying mutants with augmented or restricted PHO1 activity, our research opens avenues for the development of crop varieties with potentially higher starch content. This could lead to improvements in crop yield and quality. Furthermore, elucidating how the phosphorylation status of PHO1 influences PHO1 activity could pave the way for the creation of crop varieties with altered starch properties.

References should be presented in a consistent style. More than one author and et al would be needed. In addition, reference style manager misstates the surname in the abbreviation: e.g. ref. 1 Cori, G.T.C.F.C.S., G

line 647 Plos One

line 596 Int J Mol Sci instead of International Journal of Molecular Sciences etc..

Author response: Thank you to underline this deficiency. The overall references style have been updated according to MDPI journal standard. Specified Endnote reference style package for MDPI is used. If there are still some mistake can be updated in single click.

Minor editing of English language required

Author response: Thank you for your thoughtful recommendations. We have made significant improvements to the language of our manuscript, and it has been thoroughly reviewed by a native English speaker. Furthermore, grammar mistakes have also been checked and corrected. We trust that the quality of the language now meets the necessary standards, and we anticipate that it will not present any further issues.

We would like to extend our heartfelt gratitude once again for your meticulous review and insightful suggestions. Your point-to-point reading and careful observations have been instrumental in enhancing the quality of our manuscript. Such thorough feedback reflects your dedication to the advancement of scientific literature, and we deeply appreciate the time and effort you have invested in reviewing our work.

We look forward to hearing from you regarding our submission. We would be glad to respond to any further questions and comments that you may have.

Thanks!

This manuscript is a resubmission of an earlier submission. The following is a list of the peer review reports and author responses from that submission.

Round 1

Reviewer 1 Report

Review of

 Site-directed Mutations at Phosphorylation Sites in Zea mays 2 PHO1 Reveal Modulation of Enzymatic Activity by Phosphory-3 lation at S566 in the L80 Region

The article needs to be edited for English and several typos/errors are present in the text.

2.1.3. Identification of phosphorylated protein (PHO1) 127

To investigate the phosphorylation of PHO1, antibodies against the N-terminus and  L80 domain phosphorylation sites were prepared.

Are these phospho-specific antibodies? There is no mention of these AB in the M+M or their characterization. Figure 2 legends indicates use of the ZmPHO1 antibody (I assume this is the one described in M+M?)

Protein extracts from seeds at 15, 20, 129 and 25 DAP were chosen for analysis, as the expression level of PHO1 was found to be  highest at these stages. The protein extracts were phosphorylated before undergoing SDS- PAGE.

The protein extracts were phosphorylated before undergoing SDS- PAGE. What does this mean?

WB analysis of the SDS-PAGE results revealed two bands for PHO1 at each DAP stage, with the higher bands likely representing phosphorylated PHO1

The section, 2.1.4 Phosphorylation sites prediction, is meaningless.

Section 2.1.5. iTRAQ Identification of ZmPHO1 Phosphorylation Sites, describes 2 phosphorylation sites on PHO1, but these is no description of this in the M+M.

Figure 4 is of no value and could be described only or made supplementary.

For figure 6, the panels are labelled A,B, D and D.

The text describing figure 6 says: PHO1 were pre-treated with ATP before enzyme assays, while the figure says ATP + protein kinase and the legend goes back to ATP only!!. What does treating with ATP mean?

M+M

Anti-rabbit antibodies (ZmPHO1) were generated according to the protocol described in [27].

Like ref 27, they should briefly describe the method here, then provide results showing us the antibody only detects PHO1.

15 μl ZmPHO1 antibody was added with (Anti-PHO1) 1:1000 dilution in blocking buffer, then incubated for 1 hour on rotator at 366 room temperature. The gel membrane was washed three times for 10 minutes each in 1 % TBST.

15 uL of what, crude serum or affinity pure AB?

It is average, but okay.

Reviewer 2 Report

The authors study an essential enzyme for starch metabolism regarding their regulation by phosphorylation. The authors made punctual mutations in a protein region predicted as the phosphorylation target. They found that these mutations do not affect their location or expression but their activity since this can be linked to energy metabolism and ATP availability.

The authors did not make spectacular discoveries, but their work contributes to our acknowledgment of these critical enzymes for starch metabolism.

There are some improvements the authors could make in their manuscript to be more comprehensible. For instance, in Figure 2, the author can discuss and write the KDa of the proteins phosphorylated and no phosphorylated in the image. Line 111, maybe, should be late, and no lateral. Figure 6, double panel D and lack of panel C.

Reviewer 3 Report

Review of manuscript by Shoaib et al. (Site-directed Mutations at Phosphorylation Sites in Zea mays PHO1 Reveal Modulation of Enzymatic Activity by Phosphorylation at S566 in the L80 Region)

The manuscript by Shoaib et al. describes a study with maize (Zea mays L.) plastidial starch phosphorylase (Pho1) which attempts to elucidate the role of two putative phosphorylation sites (Ser-69 and Ser-566). Pho1 is a member of the alpha-glucan phosphorylase family of enzymes and plays a role in starch metabolism. Studies have shown that the enzyme can be phosphorylated, but the precise biological role of protein phosphorylation of Pho1 is not clear. The study examines the expression of Pho1 in maize endosperm, and analyzes the effects of removing and replacing the serine (Ser) residues at 2 potential sites using site directed mutagenesis using recombinant Pho1 proteins.

The paper lacks a cohesive thread of argument and is very disjointed. There are many questions surrounding the experimental approaches and results that need addressing and explaining properly. The first 6 lines of the abstract are superfluous and should be removed. Many statements in the text are not backed up by cite-able evidence from the literature.

1. What is the purpose of the gene and protein expression study in Figure 1? A number of studies have shown expression of Pho1 during endosperm development, so the experiment is not novel, but in any case, it has little or no bearing on the subsequent experiments.

2. Western blot data in Fig. 2 is not clear. The band shifts claimed to be associated with phosphorylation are very dramatic; in many cases with phospho-proteins if shifts are observed, they are typically very small shifts; there was no discussion of this unusual observation. How were the antibodies generated that can identify phosphorylated and dephosphorylated forms of Pho1? Are they peptide, site-specific antibodies? If so, which site (s)?

3. There is no explanation given as to how the two phosphorylation sites (Ser-69 and Ser-566) were identified. Were they predicted or actually identified experimentally? If the latter, then the data needs to be included in the paper. The results mention iTRAQ but there is no mention of this and associate methodology the Materials and Methods section.

4. Lines 162-170 are speculative and belong in the Discussion.

5. The suggestion that dephosphorylation can enhance localized expression of Pho1 is not borne out by any data presented.

6. Figure 4 is unnecessary as it merely shows the results of the site directed mutagenesis experiments, and these are standard procedure nowadays.

7. No rationale is given as to why the 2 potential sites for phosphorylation were analyzed.

8. Description of the expression of the recombinant proteins (lines 227-237) is not warranted in the text; it should be mentioned briefly in the methods section, if at all.

9. Why was the assay for Pho1 carried out only in the forward direction?

10. Important details regarding the phosphorylation of the recombinant Pho1 proteins is lacking. How were the purified proteins phosphorylated? There is no mention or explanation given in the methods. What was the source of protein kinase? Protein kinase is mentioned in Fig 6, but nowhere else. Fig 6 A and B are not necessary in the manuscript, at most they should be in a supplementary file.

Needs attantion (see my comments)